# Retrospectively Quantified T2 Improves Detection of Clinically Significant Peripheral Zone Prostate Cancer

**DOI:** 10.3390/cancers17030381

**Published:** 2025-01-24

**Authors:** Haoran Sun, Lixia Wang, Timothy Daskivich, Shihan Qiu, Hsu-Lei Lee, Chang Gao, Rola Saouaf, Eric Lo, Alessandro D’Agnolo, Hyung Kim, Debiao Li, Yibin Xie

**Affiliations:** 1Biomedical Imaging Research Institute, Cedars-Sinai Medical Center, Los Angeles, CA 90048, USA; haoran.sun@cshs.org (H.S.); lixia.wang@cshs.org (L.W.); sqiu19@g.ucla.edu (S.Q.); hsu-lei.lee@cshs.org (H.-L.L.); 2Department of Bioengineering, University of California Los Angeles, Los Angeles, CA 90095, USA; 3Urology, Cedars-Sinai Medical Center, Los Angeles, CA 90048, USA; timothy.daskivich@cshs.org (T.D.); eric.lo@cshs.org (E.L.); hyung.kiml@cshs.org (H.K.); 4Siemens Medical Solutions USA, Inc., Los Angeles, CA 90048, USA; chang.gao@siemens-healthineers.com; 5Imaging, Cedars-Sinai Medical Center, Los Angeles, CA 90048, USA; rola.saouaf@cshs.org; 6Nuclear Medicine, Cedars-Sinai Medical Center, Los Angeles, CA 90048, USA; alessandro.dagnolo@cshs.org

**Keywords:** clinically significant prostate cancer, multi-parametric MRI, quantitative imaging, T2 mapping, radiomics, machine learning, peripheral zone, prediction

## Abstract

Prostate multiparametric MRI (mpMRI) is a widely used non-invasive imaging approach for prostate cancer (PCa) diagnosis. T2-weighted imaging, a key component of mpMRI, is qualitative and indispensable for radiomics analysis. Studies show that texture-based features extracted from T2-weighted images improve PCa diagnosis. In contrast, T2 mapping can represent intrinsic physical properties of the prostate and provide quantitative information in PCa diagnosis. Based on our previous work, where a deep learning network was developed to estimate T2 maps from clinically acquired T1- and T2-weighted images, this study explores the added value of the estimated T2 maps in detecting clinically significant PCa (csPCa). The developed radiomics model utilizing features from both T2-weighted images and estimated T2 maps significantly outperformed the model relying solely on T2-weighted image features, highlighting the potential of estimated T2 maps to improve csPCa prediction.

## 1. Introduction

Prostate cancer (PCa) is the most common cancer among men, with an estimated 299,010 new cases and 35,250 deaths projected in the United States for 2024 [1,2]. Current clinical guidelines indicate that accurate detection and timely treatment of clinically significant PCa (csPCa) are crucial for effective patient management and improving prognosis [3].

The traditional PCa diagnostic pathway typically involves prostate-specific antigen (PSA) testing, digital rectal examination (DRE), and transrectal ultrasound (TRUS)-guided biopsy. However, the screening has a high false positive rate and a substantial false negative rate for PCa detection [4,5]. Additionally, the invasive nature of biopsies introduces the risk of sampling errors, side effects, and complications [6,7].

To address these challenges, multiparametric MRI (mpMRI) has emerged as a non-invasive imaging tool for PCa detection and localization. Combined with the Prostate Imaging Reporting and Data System (PI-RADS) v2.1 guideline [8], mpMRI provides both anatomical and functional information, significantly improving the accuracy of csPCa diagnosis, and reducing the risks of overdiagnosis and overtreatment [9,10,11,12]. However, accurate categorization of prostate lesions using mpMRI requires extensive radiological expertise, and interobserver reproducibility remains moderate [13,14].

Radiomics analysis has gained recognition as an alternative to conventional methods for precision diagnosis and treatment [15,16]. By extracting various types of image features and applying machine learning algorithms, radiomics has been utilized to predict PCa aggressiveness [17,18,19,20]. T2-weighted images, a key component of mpMRI, provide many features useful in this process. However, T2-weighted images only offer relative intensity values, which can vary depending on imaging parameters, sites, and scanners.

Recently, T2 mapping, a quantitative imaging technique, has shown promise in objectively representing intrinsic physical properties with higher repeatability and reproducibility [21,22]. This technique reduces observer and scanner variability, offering high diagnostic accuracy in differentiating PCa, comparable to the performance of apparent diffusion coefficient (ADC) values [23,24,25,26]. Despite these advantages, T2 mapping is not yet a part of conventional clinical practice due to the need for specific imaging sequences and additional scan time.

To address these limitations, our previous work developed a deep learning network that retrospectively estimates T2 maps from conventionally acquired T1-weighted and T2-weighted images [27]. In this study, we further evaluate the added value of retrospective T2 mapping by comparing and combining the estimated T2 maps with conventional T2-weighted images in radiomics. Radiomics models were developed to differentiate peripheral zone (PZ) csPCa, utilizing the feature information provided by both T2-weighted images and estimated T2 maps.

## 2. Materials and Methods

### 2.1. Patient Population and Data Acquisition

In vivo studies were approved by the institutional review board of Cedars-Sinai Medical Center. Informed consent was obtained from all study subjects before enrollment. Between January 2018 and December 2022, biopsy-confirmed PZ lesions of PCa patients were retrospectively analyzed.

The detailed flowchart of patient inclusion and exclusion criteria is shown in Figure 1. A total of 61 patients received mpMRI. All images were acquired with 3T MRI scanners (Biograph mMR; Siemens Healthineers, Erlangen, Germany). The implemented MRI protocols are listed in Table 1. All mpMRI scans were reviewed by a 25-years experienced radiologist following the PI-RADS v2.1 guideline [8].

Both TRUS-guided 12-core systematic biopsy and MRI-targeted biopsy, localized by a prostate cylindrical coordinate system [28], were performed on every patient. PZ lesions with Gleason Score of 3 + 4 or higher were defined as csPCa, while PZ lesions with a Gleason Score of 6 were defined as clinically insignificant prostate cancer (ciPCa).

The following exclusion criteria were applied: (1) poor mpMRI quality due to extensive motion blur or strong artifacts due to metal implant, and (2) mixed with other zonal tumor such as transition zone tumor. The final data cohort comprised 76 lesions from 58 PCa patients. Patient characteristics are shown in Table 2.

### 2.2. Data Generation and Preprocessing

The estimated T2 maps were generated using a deep learning method previously developed in our study [27]. The network was trained with clinically acquired T1-weighted and T2-weighted images as inputs and T2 maps as the reference. The training dataset consisted of 25 subjects, including 17 PCa patients and 8 healthy volunteers. To prevent misalignment between the input T2-weighted image and the reference T2 map, the T2-weighted image, as one of the inputs to train the network, was derived from the multi-echo spin-echo scans used for T2 mapping. Both the training and inference T2-weighted images have the same echo time. The network structure and training strategy were consistent with the methods described in our previous work [27].

An experienced radiologist provided a slice-by-slice tumor region of interest (ROI) segmentation for both the T2-weighted image and the estimated T2 map using ITK-SNAP 3.8.0 (www.itksnap.org) [29]. All the tumor segmentations were focused on PZ lesions that were mpMRI visible and biopsy-positive (Gleason Score ≥ 6).

### 2.3. Radiomics Analysis

#### 2.3.1. Feature Extraction

For each patient, the T2-weighted images were z-score normalized, while the estimated T2 maps were directly generated by the trained network without further processing before the radiomics analysis. Figure 2 illustrates the overall pipeline of building the radiomic models for PZ csPCa detection using conventional T2-weighted images and the estimated T2 maps. Radiomics features were extracted from the ROIs of the lesions from both T2-weighted images and the estimated T2 maps using PyRadiomics 3.1.0 in Python 3.7.6, following the guideline of Image Biomarker Standardization Initiative (IBSI) [30,31]. For each image type, 107 radiomics features were extracted: 18 first-order features, 14 shape features, and 75 texture features, including Gray-level Co-occurrence Matrix (GLCM) features, Gray-level Run Length Matrix (GLRLM) features, Gray-level Size Zone Matrix (GLSZM) features, Gray Level Dependence Matrix (GLDM) features, and Neighboring Gray Tone Difference Matrix (NGTDM) features.

#### 2.3.2. Model Development

The study cohort included 76 PZ lesions, with a positive/negative ratio of approximately 2:1 (50 csPCa cases and 26 ciPCa cases). To address the class imbalance in the training dataset, the Synthetic Minority Oversampling Technique (SMOTE) was used to balance the positive and negative samples. Min–max normalization was applied to both the T2-weighted and estimated T2 map feature matrices to the range of [0, 1].

This proof-of-concept study aimed to investigate whether the radiomic features extracted from the estimated T2 map were complementary to T2-weighted image features. Three feature selection methods, Kruskal–Wallis (KW), Relief, and Recursive Feature Elimination (RFE), with five classification methods, logistic regression (LR), support vector machine (SVM), Gaussian Process (GP), least absolute shrinkage and selection operator logistic regression (LASSO-LR), and random forest (RF), were used for feature selection and development to predict the csPCa in the PZ. Given the limited dataset size, the 5-fold cross-validation scheme was applied for the model development. The area under the receiver-operator characteristic (ROC) curve (AUC) of the validation results was used to choose the top-predictive features from the best results for the combinations of feature selection and classification method. This grid-search-like approach allowed us to evaluate and compare the overall predictive ability of the selected features from both T2-weighted images and the estimated T2 maps. For T2-weighted image and the estimated T2 map, six top-predictive features for each type of image were selected independently first. Then, integrative features were selected from the combination of both the selected T2-weighted features and estimated T2 map features using the same grid-search-like approach. The processes described above were implemented with FeAture Explorer Pro (FAE, V 0.5.14) on Python (3.7.6) [32].

### 2.4. Statistical Analysis

The performance of the models developed through a 5-fold cross-validation scheme with (a) selected T2-weighted features, (b) selected T2 map features, and (c) integrative selected T2-weighted image and T2 map features were evaluated using receiver operating characteristic (ROC) curve analysis. Accuracy, sensitivity, specificity, positive predictive value (PPV), and negative predictive value (NPV) were calculated at a cutoff value that maximized the Youden index. The 95% confidence interval (CI) was derived by bootstrapping with 1000 samples. The area under the curve (AUC) calculated from the ROC curves was used to assess model performance. The DeLong test was conducted to compare the AUCs using Python 3.7.6 [33].

## 3. Results

### 3.1. Radiomics Feature Selection of T2-Weighted Images and Estimated T2 Maps

A total of 76 PZ lesions, including 50 csPCa cases and 26 ciPCa cases, were used to develop the radiomics models with 5-fold cross-validation. Using the grid-search-like method described in Section 2, the combination of KW feature selection algorithm with LR classification model was employed to pre-select the top-predictive radiomics features from T2-weighted images and the estimated T2 maps separately. As shown in the Manhattan plots in Figure 3, the KW feature selection method ranked the features from T2-weighted images and the estimated T2 maps according to their corresponding F-values, with a significance level set at *p* < 0.05.

Two groups of six features were selected according to the best validation AUCs generated by LR classification models. Table 3 shows the six selected features of the T2-weighted images and the estimated T2 maps. The selected features from the estimated T2 map are first-order features, including 10 Percentile, 90 Percentile, Interquartile Range, Mean, Mean Absolute Deviation, and Minimum. The selected features from the T2-weighted image include two texture features, GLDM Gray Level Variance and NGTDM Contrast, and four first-order features, 10 Percentile, Mean, Mean Absolute Deviation, and Robust Mean Absolute Deviation. Detailed feature definitions can be found in the PyRadiomics Documentation V3.1.0 [31]. The selected top-predictive features of csPCa have lower feature values compared to the ciPCa, with all *p*-values < 0.05.

Performance measures of the LR classifiers for each group of features were reported in Table 4 according to the ROC curves for detecting PZ csPCa cases. The model trained with T2-weighted radiomics features achieved an AUC, accuracy, sensitivity, specificity, PPV, and NPV of 0.700 (95% CI: 0.568–0.831), 0.737, 0.800, 0.615, 0.800, and 0.615, respectively, while the model trained with the estimated T2 map radiomics features achieved an AUC, accuracy, sensitivity, specificity, PPV, and NPV of 0.763 (95% CI: 0.649–0.877), 0.711, 0.640, 0.846, 0.890, and 0.550, respectively. There is a relative improvement between AUCs of 6.3% with no significant difference when performing the DeLong test to compare the AUCs of the estimated T2 map features and T2-weighted image features.

### 3.2. Integrative Model Construction

The integrative feature selection was performed on the combination of the 12 pre-selected top-predictive features from both T2-weighted image and the estimated T2 map. Using the same grid-search-like scheme, a total of nine features, including all six estimated T2 map features and three T2-weighted features, were finally constructed by using Relief selection methods with a GP classification algorithm. As highlighted in Table 3, all the six pre-selected features of the estimated T2 map are included, while the selected features from the T2-weighted image remain one GLDM texture feature, Gray Level Variance, and two first-order features, Mean Absolute Deviation and Robust Mean Absolute Deviation. In Figure 4, two true-positive predicted examples, a PZ ciPCa lesion (A) and a PZ csPCa lesion (B), were illustrated. Comparing the selected top-predictive radiomics features, the overall distribution shows a visually different pattern that the csPCa has lower feature values than the ciPCa.

The best performance of the models trained with the final selected nine top-predictive features achieved the AUC, accuracy, sensitivity, specificity, PPV, and NPV of 0.803 (95% CI: 0.694–0.913), 0.803, 0.780, 0.846, 0.907, and 0.667, respectively. As shown in Table 4 and Figure 5, compared with T2-weighted image features and estimated T2 map features only, the relative improvements of AUCs are 10.3% and 3.7%, respectively. There is a significant *p*-value of 0.048 between the integrative model AUC and the T2-weighted model AUC.

## 4. Discussion

In this study, we developed radiomic models to predict PZ csPCa using features extracted from both T2-weighted images and estimated T2 maps. Integrating selected radiomic features from these two types of images significantly enhanced csPCa prediction compared to using features from T2-weighted images alone. This demonstrates the added value of quantitative information provided by radiomic features derived from the estimated T2 maps.

Currently, prostate cancer characterization and monitoring often involves frequent biopsies. However, MRI-derived radiomic analysis is increasingly being explored as a non-invasive method for assessing PCa aggressiveness. T2-weighted images, a key component of mpMRI, provide high spatial resolution and tissue-specific contrast, which are vital for radiomics analysis. However, unlike diffusion-weighted imaging (DWI)-derived apparent diffusion coefficient (ADC) maps, which offer quantitative information, T2-weighted images are limited by their qualitative nature. The signal intensities in T2-weighted images reflect relative contrast between tissues, which can vary due to protocol parameters and radiofrequency inhomogeneity. Standardization methods, such as mean and z-score normalization, are commonly used to address this issue. More sophisticated methods use a pair of reference tissue signal intensities as a normalization scalar for T2W images to create a pseudo-T2 map [34,35]. However, these preprocessing methods may require the labor of reference tissue segmentations, and the estimated T2 varies from different reference tissue pairs.

T2 mapping, on the other hand, provides quantitative and reproducible T2 values that can directly assess prostate cancer aggressiveness [24,36]. To take advantage of this quantitative information to boost the performance of radiomics analysis, we retrospectively estimated the T2 maps without the need for additional scan time or specific sequences. In our study, the estimated T2 maps provided six first-order features that contributed to the integrative model for predicting clinically significant PCa. These first-order features describe the voxel intensity distribution within tumor regions of interest (ROIs), capturing the quantitative nature of T2 mapping. Tumors with intermediate to high-grade PCa exhibited significantly lower T2 values compared to low-grade tumors, consistent with the distribution of the selected estimated T2 map features.

DWI and ADC, as essential components of mpMRI, play a crucial role in clinical prostate cancer diagnosis. These sequences provide quantitative diffusion information that reflects prostate cancer aggressiveness, and studies have demonstrated their potential in radiomics models for PCa diagnosis [17,37,38]. However, in this study, DWI and ADC features were excluded to focus on evaluating the specific value of radiomics features extracted from the estimated T2 maps. Our dataset presented substantial image mismatch between T2-weighted/T2 maps and DWI/ADC images due to differences in acquisition protocols and spatial resolution, DWI distortions, and variations in bladder size during scanning. These mismatches would compromise the reliability of combining regional quantitative features from different modalities, a critical requirement for radiomics analysis. Nevertheless, integrating DWI and ADC features has the potential to enhance the overall predictive performance of radiomics models. In the future, addressing the image mismatch issues through advanced image registration techniques tailored for the pelvic area would enable a more comprehensive analysis that combines the image information from all mpMRI sequences.

T2-weighted images, despite their qualitative nature, offer high spatial resolution and tissue-specific contrast, making their textural features valuable for assessing PCa aggressiveness. Studies demonstrated that texture features that represent homogeneity on T2-weighted images are positively correlated with prostate cancer Gleason Score [39,40]. In our study, three radiomics features from T2-weighted images were included in the final model: one GLDM texture feature (Gray Level Variance) and two first-order features (Mean Absolute Deviation and Robust Mean Absolute Deviation). Comparing these selected feature values, the csPCa has significantly lower variance and deviation than the insignificant ones, which correlated with the previous studies that the high-grade tumors are more homogeneous compared to the low-grade tumors. Pathologically, different aggressiveness levels of prostate cancer show differences in internal cellular components, fluid contents, collagen levels, and fibromuscular stroma with other features. High-grade PCa is poorly differentiated and characterized by high cellularity and decreased extracellular space. Low-grade tumors have at least some remaining glandular structures, which preserve some intercellular space [17,41]. The final selected features from the T2-weighted image revealed that tumor homogeneity further improved the radiomics-based assessment of PCa aggressiveness.

Our final integrative radiomics model incorporated six features from the estimated T2 maps and three from T2-weighted images. The quantitative nature of the T2 maps provided valuable intensity-based information, while the qualitative T2-weighted images contributed variance-related information through both first-order and textural features. By combining both the T2-weighted features and the estimated T2 map features, the model for assessing PZ PCa aggressiveness was significantly improved compared to using T2-weighted image features alone. This reinforces the added value of the retrospectively generated T2 maps in non-invasive PCa assessment. As the estimated T2 maps were retrospectively quantified from clinically acquired T1- and T2-weighted images, this approach can be seamlessly integrated into clinical practice. The developed radiomics model can function as a decision-support tool within radiological workflows, offering objective and reproducible quantitative assessments.

Our study has several limitations. First, as mentioned, DWI and ADC were not included in the radiomics models due to the focus on evaluating the added value of the estimated T2 map features and the challenges associated with image mismatch. Second, this study only investigated clinically significant PCa in PZ lesions. Radiomic features and models were not developed or evaluated for lesions in other prostate zones. PZ tumors account for approximately 70–75% of prostate cancer cases and exhibit higher T2 value ranges for both normal tissue and lesions compared with tumors in other zones [8,22,36]. Furthermore, radiomics studies have demonstrated that features that have been selected most useful in cancer detection are different between zones [42]. To evaluate the clinical value of the quantitative information provided by the estimated T2 map, our study is currently only focusing on PZ lesions. Third, our study cohort consisted of 76 cases from a single institution, using consistent mpMRI protocol on the same scanner. Due to the small sample size, we employed a 5-fold cross-validation scheme without a separate testing dataset, and the added value of the estimated T2 map radiomics features was assessed using validation area under the curve (AUC) scores. Future work should be dedicated to developing separate models that work for lesion assessment in all three zones, respectively. DWI, ADC map, and DCE should be included after addressing image mismatch in the future. The generalizability and reproducibility of the selected radiomics features should be further investigated and validated by larger external datasets.

## 5. Conclusions

This study demonstrated that the deep learning network-estimated T2 map can improve the detection of csPCa in the PZ using radiomics models. The combination of features derived from both conventional qualitative T2-weighted images and retrospectively estimated T2 maps significantly improved prediction accuracy compared to relying solely on T2-weighted images. The radiomics features extracted from estimated T2 maps provided additional quantitative information, contributing to patient risk stratification. These findings underscore the potential of retrospective T2 mapping as a valuable tool in the precision diagnosis and management of prostate cancer.

## Figures and Tables

**Figure 1 cancers-17-00381-f001:**
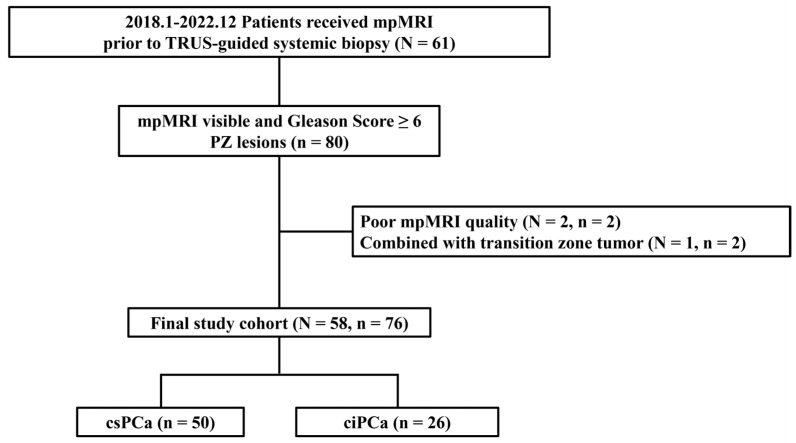
Patient inclusion/exclusion flowchart. A 58 prostate cancer (PCa) patients dataset leading to 76 peripheral zone (PZ) lesions were divided into two classes: 50 clinically significant PCa (csPCa) lesions and 26 clinically insignificant prostate cancer (ciPCa) lesions. Patient number is represented by N; lesion number is represented by n.

**Figure 2 cancers-17-00381-f002:**
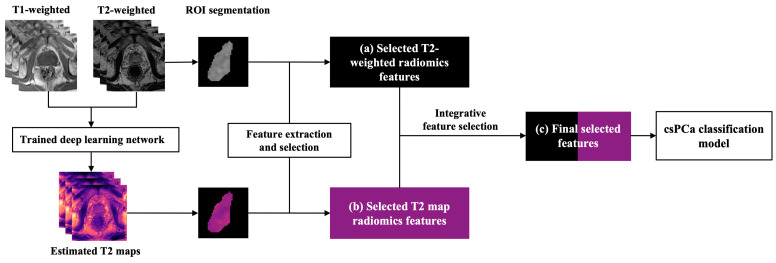
The workflow for building the radiomic models for csPCa detection using estimated T2 maps and conventional T2-weighted images. The estimated T2 maps were generated using a trained deep learning network from T1-weighted and T2-weighted images. Radiomics features of the lesion ROIs were extracted and selected independently from both T2-weighted images and the estimated T2 maps. An integrative feature selection was performed after combining the selected T2-weighted and T2 map features. Finally, csPCa classification models were developed using the three groups of radiomics features: (**a**) selected T2-weighted features, (**b**) selected T2 map features, (**c**) integrative selected T2-weighted and T2 map features.

**Figure 3 cancers-17-00381-f003:**
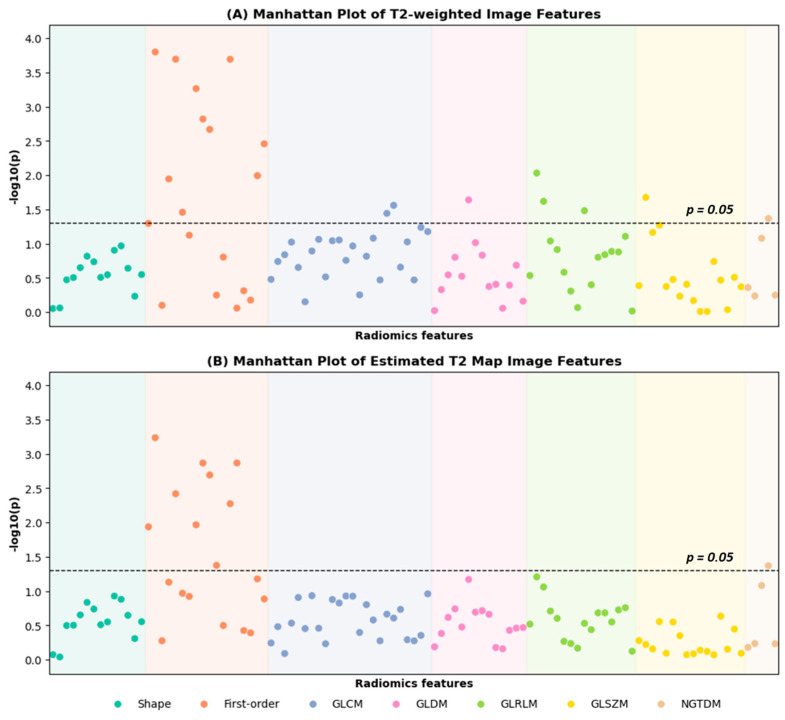
Manhattan plots of all extracted radiomics features from (**A**) T2-weighted image and (**B**) estimated T2 map. The Manhattan plot shows the *p*-values for all radiomic features between clinically significant PCa and clinically insignificant PCa. Radiomic features are lined up on the x-axis, whereas the -log10 (*p*) values are plotted on the y-axis. The dashed line indicates the *p*-value of 0.05. The eighteen T2-weighted image radiomic parameters and ten estimated T2 map parameters above the dashed line were considered statistically significant.

**Figure 4 cancers-17-00381-f004:**
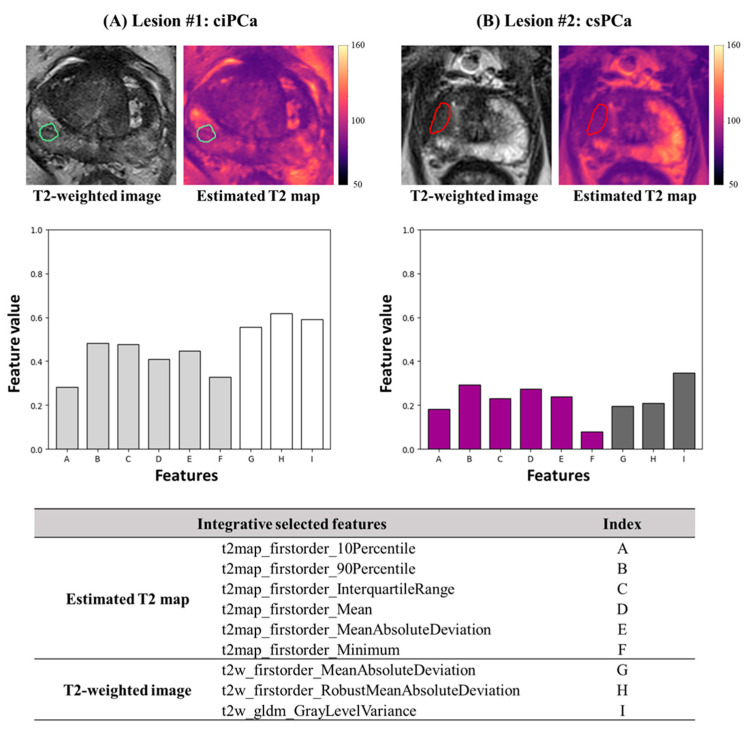
Visualization of T2-weighted images and the estimated T2 maps, and the value of the integrative selected top-predictive features for two true-positive predicted lesions. (**A**) Depicts a PZ ciPCa lesion, with the lesion ROI outlined in green on both T2-weighted image and estimated T2 map ([ms]) of the same slice. The histogram shows the selected feature values, where light gray bars represent features extracted from the estimated T2 map and white bars represent features from the T2-weighted image. (**B**) Shows a PZ csPCa lesion, with the ROI outlined in red. The purple bars represent features from the estimated T2 map; the dark gray bars represent features from T2-weighted images. The features are indexed in the table below, corresponding to the x-axis of the histogram.

**Figure 5 cancers-17-00381-f005:**
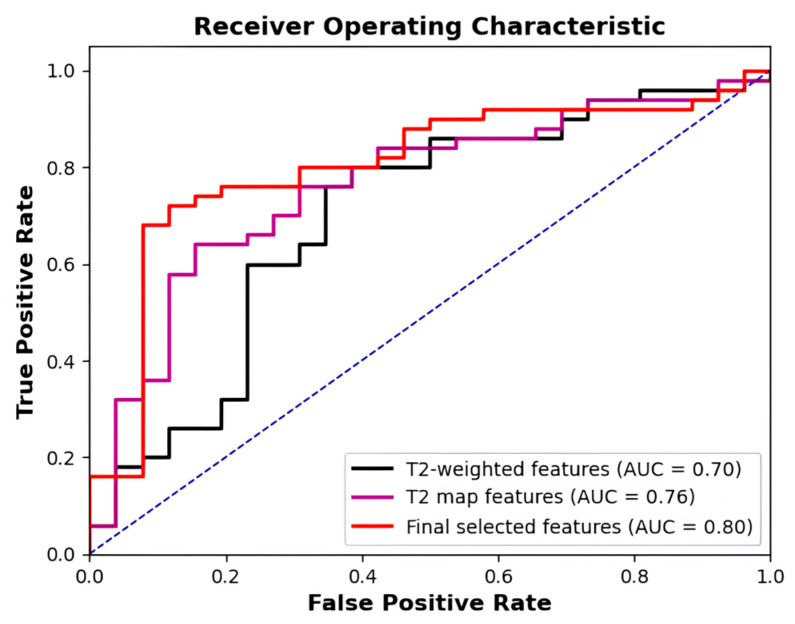
Receiver operator characteristic (ROC) curves of radiomics models utilizing the selected T2-weighted features (black), selected T2 map features (purple), and the integrative final selected T2-weighted and T2 map features (red). The AUCs of the validation results were 0.700, 0.763, and 0.803, respectively.

**Table 1 cancers-17-00381-t001:** Summary of protocol parameters for the mpMRI sequences of the prostate.

	T1w (FLASH)	T2w (TSE)	DWI (EPI)	DCE (GRE)
TE (ms)	2.03	132	95	1.07
TR (ms)	277	4000	6500	3.02
Flip angle (°)	65	158	90	10
# Slices	45	30	29	31
Thickness (mm)	6	3	3	3
Resolution (mm^2^)	1.125 × 1.125	0.63 × 0.63	0.781 × 0.781	1.250 × 1.250
FOV (mm^2^)	360 × 247.5	160 × 160	200 × 200	160 × 160
Temporal Resolution (s)	/	/	/	20
*b*-value (s/mm^2^)	/	/	50, 800, 1400	/
Scan time (min)	0.5	4.5	6.4	8.2

**Table 2 cancers-17-00381-t002:** Clinical characteristics of the patients included in the study cohort.

Characteristics	(Total Patients N = 58, Total Lesions n = 76)
Age (yr), median {IQR}	69 {63.5–73}
PSA (ng/mL), median {IQR}	6.2 {5.1–7.3}
PSAD (ng/mL^^2^), median {IQR}	0.16 {0.09–0.25}
Prostate volume (cc), median {IQR}	41.78 {26.77–58.36}
**Gleason Score, n {%}**	
3 + 3	26 {34.2}
3 + 4	34 {44.7}
4 + 3	7 {9.2}
≥4 + 4	9 {11.8}
**PI-RADS, n {%}**	
1	0 {0}
2	0 {0}
3	6 {7.9}
4	47 {61.8}
5	23 {30.3}

**Table 3 cancers-17-00381-t003:** Summary of the value of the selected features from T2-weighted image and the estimated T2 map for csPCa and ciPCa. The highlighted lines indicate the final features selected for constructing the integrative model.

T2-Weighted Features	csPCa	ciPCa	*p*-Value
firstorder_90Percentile	0.44 ± 0.14	0.58 ± 0.15	<0.001 ***
firstorder_MeanAbsoluteDeviation	0.32 ± 0.17	0.48 ± 0.18	<0.001 ***
firstorder_RobustMeanAbsoluteDeviation	0.31 ± 0.18	0.49 ± 0.21	<0.001 ***
firstorder_Mean	0.46 ± 0.15	0.58 ± 0.17	<0.01 **
gldm_GrayLevelVariance	0.30 ± 0.19	0.41 ± 0.19	0.02 *
ngtdm_Contrast	0.20 ± 0.17	0.29 ± 0.16	0.04 *
Estimated T2 Map Features	csPCa	ciPCa	*p*-Value
firstorder_90Percentile	0.40 ± 0.15	0.53 ± 0.16	<0.001 ***
firstorder_InterquartileRange	0.25 ± 0.15	0.36 ± 0.14	<0.01 **
firstorder_Mean	0.38 ± 0.15	0.51 ± 0.16	<0.01 **
firstorder_10Percentile	0.32 ± 0.16	0.43 ± 0.19	0.01 *
firstorder_MeanAbsoluteDeviation	0.29 ± 0.16	0.38 ± 0.13	0.01 *
firstorder_Minimum	0.31 ± 0.23	0.43 ± 0.22	0.04 *

* *p* < 0.05; ** *p* < 0.01; *** *p* < 0.001.

**Table 4 cancers-17-00381-t004:** Comparisons of the prediction performance between radiomics models utilizing the selected T2-weighted image features, selected estimated T2 map features, and the integrative final selected T2-weighted and T2 map features. *p*-values were generated from DeLong tests to compare the AUCs of estimated T2 map model and the integrative model with the T2-weighted model.

	AUC	95% CIs	ACC	SEN	SPE	PPV	NPV	*p*-Value
T2-weighted image	0.700	[0.568–0.831]	0.737	0.800	0.615	0.800	0.615	/
Estimated T2 map	0.763	[0.649–0.877]	0.711	0.640	0.846	0.890	0.550	0.260
T2 weighted + Estimated T2 map	0.803	[0.694–0.913]	0.803	0.780	0.846	0.907	0.667	0.043 *

* *p* < 0.05.

## Data Availability

The prostate data presented in this study are available on request from the corresponding authors upon reasonable request.

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
