# Peer review of "Retrospectively Quantified T2 Improves Detection of Clinically Significant Peripheral Zone Prostate Cancer"

_cancers, 2025, doi:10.3390/cancers17030381_

Round 1
Reviewer 1 Report
Comments and Suggestions for Authors
In this study, the authors focused on the added value of retrospective T2 mapping by comparing and combining the estimated T2 maps with conventional T2-weighted images in radiomics.
The retrospective single-center analysis comprised 61 consecutive patients (50 csPCa lesions and 26 ciPCa lesions). These were predominantly low- or intermediate (6 or 7 (3+4)) and PIRADS 4 lesions that were enrolled. Please comment on that further, as mpMRI efficacy is mainly associated with the aggressivenes of the disease. How it may affect your model accuracy?
Interesingly, no features from DWI and ADC were included in the development of the radiomics models. Please develop this issue furhter in the discussion rather then just mention in the limitations.
As you stated that the model improves the prediction of peripheral zone csPCa aggressiveness, please futher comment how it can be implemented in clinical practice?
Reviewer 2 Report
Comments and Suggestions for Authors
The authors examined the value of estimated T2 maps produced by a deep-learning network in prostate multiparametric MRI (mpMRI) for detecting clinically significant prostate cancer (csPCa).
Some comments are listed below.
1. The authors could consider using the developed radiomics model of mpMRI +PSA or other clinical characteristics to increase the accuracy in the diagnosis of csPCa.
2. In the discussion, the authors could compare the advantages and disadvantages of the developed radiomics model of mpMRI with PET MRI.
3. Provide the full name of the abbreviation in the legends of figures and tables.
4. Reference 5 is old. Are there any additional papers that support the authors’ conclusion?
